# Pharmacokinetic Alteration of Paclitaxel by Ferulic Acid Derivative

**DOI:** 10.3390/pharmaceutics11110593

**Published:** 2019-11-09

**Authors:** Jaeok Lee, Song Wha Chae, LianJi Ma, So Yeon Lim, Sarah Alnajjar, Hea-Young Park Choo, Hwa Jeong Lee, Sandy Jeong Rhie

**Affiliations:** 1College of Pharmacy and Graduate School of Pharmaceutical Sciences, Ewha Womans University, Seoul 03760, Korea; leejo19@ewha.ac.kr (J.L.); yuki1226@naver.com (S.W.C.); malianji0215@163.com (L.M.); deersy@naver.com (S.Y.L.); hypark@ewha.ac.kr (H.-Y.P.C.); 2College of Pharmacy and Division of Life & Pharmaceutical Sciences, Ewha Womans University, Seoul 03760, Korea; Najjar_sara@yahoo.com

**Keywords:** ferulic acid derivatives, P-glycoprotein, elimination, pharmacokinetics, bioavailability, paclitaxel

## Abstract

P-glycoprotein (P-gp) is known to be involved in multidrug resistance (MDR) and modulation of pharmacokinetic (PK) profiles of substrate drugs. Here, we studied the effects of synthesized ferulic acid (FA) derivatives on P-gp function in vitro and examined PK alteration of paclitaxel (PTX), a well-known P-gp substrate drug by the derivative. Compound **5c**, the FA derivative chosen as a significant P-gp inhibitor among eight FA candidates by in vitro results, increased PTX AUC_inf_ as much as twofold versus the control by reducing PTX elimination in rats. These results suggest that FA derivative can increase PTX bioavailability by inhibiting P-gp existing in eliminating organs.

## 1. Introduction

P-glycoprotein (P-gp) is a membrane efflux transporter that is a member of the ATP-binding cassette (ABC) superfamily. This transporter is widely distributed in normal organs including the brain [1], liver [2], intestine [3] and kidney [4] to protect our body from exogenous compounds including toxins by extruding them out of the body organs. P-gp is associated with a multidrug resistance (MDR) phenomenon which makes many potent anticancer drugs ineffective and is one of the reasons responsible for the failure of cancer chemotherapy [5,6,7]. Therefore, P-gp inhibitors can be used to overcome MDR. Moreover, this concept is extended to improve the absorption of orally delivered medications by inhibiting P-gp located in the gastrointestinal (GI) tract [8,9,10].

Ferulic acid (FA) is one of the most ubiquitous phenolic compounds in plants, and the organic compound and its derivatives possess multiple physiological and pharmaceutical functions such as antioxidant, antimicrobial, anti-inflammatory, antithrombosis and anticancer activities [11]. In the anticancer effect, it has been reported that FA and its derivatives induce cell cycle arrest, autophagy, apoptosis and cytotoxicity in several cancer cells such as leukemia, breast cancer, cervical cancer, etc. [12,13,14]. However, the effect of FA and its derivatives on P-gp modulation have not been investigated well. Only a few studies have assessed recently the effect of FA on the reversal of the P-gp-mediated MDR in cancer [15,16]. Therefore, we synthesized several FA derivatives in the study and evaluated their function on P-gp in vitro and in vivo systems.

## 2. Materials and Methods

### 2.1. Materials

Ferulic acid, sulforhodamine B (SRB), (+/−)-verapamil hydrochloride and other reagents for synthesis were purchased from Sigma-Aldrich (St. Louis, MO, USA). [^3^H]-daumnomycin (16 Ci/mmol) was supplied by PerkinElmer Life and Analytical Sciences (Boston, MA, USA). The BD Gentest^TM^ ATPase assay kit and human P-glycoprotein membranes were commercially provided from BD Bioscience (Woburn, MA, USA). Paclitaxel (PTX) was obtained from Samyang Genex (Daejeon, Korea). All reagents and solvents were supplied as synthesis grades, cell culture grades and HPLC analysis grades. Melting points were measured on an electro-thermal digital melting point (Buchi, Essen, Germany) without calibration. ^1^H NMR spectra were recorded on Varian NMR AS and Varian Unity Inova 400 MHz NMR spectrometers. Chemical shifts were reported in parts per million (d) units relative to the solvent peak. The ^1^H NMR data were reported as peak multiplicities: s for singlet; d for doublet; t for triplet; and m for multiplet. Coupling constants were recorded in hertz. Mass spectra data was obtained on an Agilent 6220 Accurate-Mass time-of-flight liquid chromatography/mass spectrometry (TOF LC/MS). All the synthesized compounds have been registered as a patent (KR10-2013-0099676) by Prof. Hea-Young Park Choo and Ms So Yeon Lim since 2013.

### 2.2. Synthesis of FA Derivatives

#### 2.2.1. Methylation of Ferulic Acid

Ferulic acid 2 g (10.3 mmol) was dissolved in 40 mL of methanol and 1.2 mL of concentrated sulfuric acid (H_2_SO_4_) was added, and heated under reflux overnight. Then, 5% NaHCO_3_ was added to make alkaline solution (pH 10) and extracted with diethyl ether. The solution was dried over MgSO_4_ and the solvent was removed in vacuo.

#### 2.2.2. General Procedure for the Synthesis of Esters

To the methyl ferulate (0.5 g, 2.4 mmol) in dry acetone (6 mL) solution, pottassium carbonate (K_2_CO_3_) (0.4 g, 2.9 mmol) was added; 1-Iodopropan (0.26 mL, 2.9 mmol) was added and the reaction mixture was heated at 60 °C for 3 h. After cooling, the reaction mixture was extracted with diethyl ether, dried over MgSO_4_ and the solvent was removed in vacuo. Column chromatography (hexane: ethylacetate = 1:1) afforded the desired product.

#### 2.2.3. General Procedure for Hydrolysis

The ester obtained from the procedure 2.2.2 (0.5 mmol) was dissolved in acetone (6 mL), and 2 N NaOH (6 mL) was added and the reaction mixture was heated to 90 °C for 2 h. The reaction mixture was extracted with diethyl ether. The water layer was acidified with 10% HCl solution (pH 2) and extracted with diethyl ether. The organic solution was dried over MgSO_4_ and the solvent was removed in vacuo to obtain the desired product.

#### 2.2.4. General Synthesis Amides

Various amines (0.23 mmol) and PyBOP (benzotrial-1-yl-oxytripyrrolidinophosphonium hexafluorophosphate) (0.12 g, 0.23 mmol) was added to the prepared (E)-3-(3-methoxy-4-substituted phenyl)acrylic acid (0.21 mmol) in dimethylformamide (5 mL). After stirring, i-Pr_2_NEt (0.007 mL, 0.46 mmol) was added and stirred at room temperature for 16 h. The reaction mixture was extracted with diethyl ether, dried over MgSO_4_ and the solvent was removed in vacuo.

##### Synthesis of (E)-3-(3-methoxy-4-propoxyphenyl)-1-(piperidin-1-yl)prop-2-en-1-one (Compound **5a**)

Piperidine (0.023 mL, 0.23 mmol) was employed as amine. Pale yellow solid (47%) was obtained.; mp 91–92 °C; ^1^H NMR (400 MHz, CDCl_3_) δ 7.59 (1H, *d*, *J* = 15.2 Hz), 7.08 (1H, *d*, *J* = 6.4Hz, 7.03 (1H, s), 6.85 (1H, *d*, *J* = 8.4 Hz), 6.75 (1H, *d*, *J* = 15.2 Hz), 4.00 (2H, *t*, *J* = 6.8 Hz), 3.90 (3H, s), 3.65–3.60 (4H, m), 1.90–1.85 (2H, m), 1.69–1.54 (6H, m), 1.04 (3H, *t*, *J* = 7.6 Hz).

##### Synthesis of (E)-3-(4-isobutoxy-3-methoxyphenyl)-1-(piperidin-1-yl)prop-2-en-1-one (Compound **5b**)

Piperidine (0.023 mL, 0.23 mmol) was employed as amine. Yellow solid (83%) was obtained.; mp 96–97 °C; ^1^H NMR (400 MHz, CDCl_3_) δ 7.58 (1H, *d*, *J* = 15.2 Hz), 7.07(1H, *d*, *J* = 6.8 Hz), 7.03 (1H, s), 6.84 (1H, *d*, *J* = 8.4 Hz), 6.75 (1H, *d*, *J* = 15.2 Hz), 3.90 (3H, s), 3.79 (2H, *d*, *J* = 6.8 Hz), 3.65–3.60 (4H, m), 2.20–2.14 (1H, m), 1.68–1.53 (6H, m), 1.04 (3H, s).

##### Synthesis of (E)-3-(4-benzyloxy-3-methoxyphenyl)-1-(piperidin-1-yl)prop-2-en-1-one (Compound **5c**) [17]

##### Synthesis of (E)-3-(3-methoxy-4-propoxyphenyl)-*N*,*N*-dimethylacrylamide (Compound **5d**)

Dimethylamine HCl (0.02 g, 0.23 mmol) was employed as amine. Yellow solid (23%) was obtained.; mp 108–109 °C; ^1^H NMR (400 MHz, CDCl_3_) δ 7.61 (1H, *d*, *J* = 15.2 Hz), 7.098 (1H, *d*, *J* = 6.4 Hz), 7.04 (1H, s), 6.86 (1H, *d*, *J* = 8.4 Hz), 6.74 (1H, *d*, *J* = 15.6 Hz), 4.01 (2H, *t*, *J* = 6.8 Hz), 3.91 (3H, s), 3.18 (3H, s), 3.07 (3H, s), 1.91–1.85 (2H, m), 1.05 (3H, *t*, *J* = 7.9 Hz).

##### Synthesis of (E)-3-(4-isobutoxy-3-methoxyphenyl)-*N*,*N*-dimethylacrylamide (Compound **5e**)

Dimethylamine HCl (0.018 g, 0.22 mmol) was employed as amine. White solid (46%) was obtained.; mp 109–111 °C; ^1^H NMR (400 MHz, CDCl_3_) δ 7.61 (1H, *d*, *J* = 15.6 Hz), 7.09 (1H, *d*, *J* = 6.0 Hz), 7.04 (1H, s), 6.84 (1H, *d*, *J* = 8.4 Hz), 6.74 (1H, *d*, *J* = 15.6 Hz), 3.90 (3H, s), 3.80 (2H, *d*, *J* = 6.8 Hz), 3.18 (3H, s), 3.10 (3H, s), 2.19–2.16 (1H, m), 1.04 (3H, s), 1.03 (3H, s).

##### Synthesis of (E)-*N*-benzyl-3-(3-methoxy-4-propoxyphenyl)-N-methylacrylamide (Compound **5f**)

*N*-methyl benzylamine (0.028 mL, 0.23 mmol) was employed as amine. Yellow oil (56%) was obtained.; ^1^H NMR (400 MHz, CDCl_3_) δ 7.71 (1H, *d*, *J* = 15.2 Hz), 7.38-7.24 (5H, m), 7.12-7.03 (1H, m), 6.95 (1H, s), 6.87-6.71(2H, m), 4.71 (1H, s), 4.00 (2H, *d*, *J* = 8.4 Hz), 3.88 (3H, s), 3.08 (3H, s), 1.89–1.85 (2H, m), 1.05 (3H, *t*, *J* = 7.6 Hz).

##### Synthesis of (E)-3-(4-benzyloxy-3-methoxyphenyl)-*N*,*N*-dimethylacrylamide (Compound **5g**)

Dimethylamine HCl (0.015 g, 0.19 mmol) was employed as amine. Pale yellow solid (67%) was obtained.; mp 135–136 °C; ^1^H NMR (400 MHz, CDCl_3_) δ 7.60 (1H, *d*, *J* = 15.2 Hz), 7.45-7.31 (5H, m), 7.06–7.04 (2H, m), 6.86 (1H, *d*, *J* = 8.8 Hz), 6.74 (1H, *d*, *J* = 15.6 Hz), 5.86 (2H, s), 3.93 (3H, s), 3.17 (3H, s), 3.07 (3H, s).

##### Synthesis of (E)-1-(4-hydroxy-4-phenylpiperidin-1-yl)-3-(3-methoxy-4-propoxyphenyl) prop-2-en-1-one (Compound **5h**)

4-Phenylpiperidine-4-ol (0.041 g, 0.23 mmol) was employed as amine. Pale yellow solid (52%) was obtained.; mp 163–164 °C; ^1^H NMR (400 MHz, CDCl_3_) δ 7.64 (1H, *d*, *J* = 15.6 Hz), 7.48 (2H, *d*, *J* = 8.8 Hz), 7.38 (2H, *t*, *J* = 11.6 Hz), 7.30 (1H, *d*, *J* = 5.6 Hz), 7.10 (1H, *d*, *J* = 8.8 Hz), 7.05 (1H, s), 6.86 (1H, *d*, *J* = 8.4 Hz), 6.80 (1H, *d*, *J* = 15.2 Hz), 4.70 (1H, s), 4.13–4.01 (3H, m), 3.91 (3H, s), 3.67 (1H, s), 3.24 (1H, s), 2.10–2.04 (2H, m), 2.04–1.80 (4H, m), 1.05 (3H, *t*, *J* = 7.6 Hz).

### 2.3. Cytotoxicity Studies in P-gp Overexpressed Cells

The effect of eight FA derivatives on cytotoxicity was studied in P-gp overexpressed human breast cancer cells (MCF-7/ADR) using the SRB assay [18]. The details of the cell culture condition and the assay method were presented in our previous reports [10,19]. Verapamil (VER, 100 μM), one of the P-gp inhibitors, was used as a positive control. The half maximal inhibitory concentration (IC_50_) values were calculated with Table Curve2D^®^ version 5.01 software (Systat Software Inc., San Jose, CA, USA). The assay was performed in triplicate.

### 2.4. [^3^H]-Daunomycin Accumulation and Efflux Studies

Among eight FA derivatives, compounds **5c**, **5f**, **5g** and **5h** (100 μM) were selected for [^3^H]-daunomycin (DNM) accumulation and efflux studies based on cytotoxicity results. The methods for [^3^H]-DNM accumulation and efflux studies were reported previously [10,19]. VER (100 μM) was used as a positive control. The experiments were performed in triplicate.

### 2.5. Human P-glycoprotein ATPase Activity Assay

The effects of compounds **5c**, **5f**, **5g** and **5h** on P-gp ATPase activity at different concentrations (20, 50 and 100 μM) were examined in human P-gp membranes using an ATPase assay kit according to the method reported previously [10,19]. VER was used as a P-gp inhibitor and an ATPase stimulator. The ATPase activities were expressed as the rate of phosphate release per milligram of membrane protein and converted to the relative ratio versus the control. This assay was performed in duplicate.

### 2.6. Pharmacokinetic Study

The pharmacokinetic (PK) study was performed using male Sprague-Dawley rats (6 weeks old and 200 g–235 g) commercially available from Orient Bio (Seongnam, Korea) [10,19]. All animal procedures were approved by the Institutional Animal Care and Use Committee of Ewha Womans University (No. 2012-01-019, approved on 3 April 2012), Republic of Korea.

Among eight FA derivatives, compound **5c** ((E)-3-(4-(benyloxy-3-methoxyphenyl)-1-(piperidin-1-yl)prop-2-en-1-one) was chosen to examine the effect on PTX pharmacokinetics because it was found to be most effective in inhibiting P-gp function in vitro. Taxol formulation (Cremophor^®^ EL, anhydrous ethanol and isotonic saline (1/1/4, *v*/*v*/*v*)) was used to dissolve the PTX and compound **5c** to a final concentration of 2 mg/mL [10,19]. All were prepared immediately prior to use.

Rats were divided into the following 4 groups: Oral administration (PO) of PTX (25 mg/kg) alone and co-administration of PTX (25 mg/kg) with compound **5c** at three different doses (0.5, 2 and 5 mg/kg). Each group had 5 rats except for a group of 2 mg/kg dose (*n* = 4). The blood samples (0.2 mL) were collected from the common carotid artery at 0, 0.25, 0.5, 1, 2, 3, 4, 6, 8, 10 and 24 h.

PTX concentrations in rat plasma were analyzed by Agilent HP1100 series system using a Capcell-pak C_18_ MG120 column (3 mm × 250 mm, 5 µm, Shiseido, Tokyo, Japan). The samples were eluted with mobile phase composed of acetonitrile and 0.1% phosphoric acid (1/1, *v*/*v*) at a flow rate of 0.5 mL/min for 30 min and detected at a wavelength of 227 nm. The HPLC-UV method was validated by FDA guidelines as follows: Specificity (PTX: 14.8 min; internal standard: 26.7 min), linearity over the concentration range between 0.01 and 10 µg/mL, lower limit of quantification (0.01 µg/mL) and intra-/inter-day precision and accuracy (2.4–9.1% and 92.1–102.7%, respectively) [20]. All other details for sample preparation and quantification of PTX concentrations in rat plasma were reported previously [8,9,10].

### 2.7. Pharmacokinetic Analysis

The following PK parameters of PTX after oral single dosing to rats were estimated by non-compartmental analysis using WinNonlin^®^ Professional version 5.2 software (Pharsight Corporation, Mountain View, CA, USA): The area under the plasma concentration-time curve from 0 h to infinity (AUC_inf_), elimination half-life (t_1/2_), apparent volume of distribution after oral single dosing (Vz/F), oral clearance (Cl/F), maximum plasma concentration (C_max_) and the time required to reach C_max_ (T_max_). The relative bioavailability (RB, %) of PTX was calculated with the following formula:
$$RB\;\left(\;\%\;\right)=\frac{AUCinf\;po\;co-administraion}{AUCinf\;po\;control}\times 100$$

AUC_inf_ po control is the AUC_inf_ obtained from oral single administration of PTX alone, and AUC_inf_ po co-administration is the AUC_inf_ obtained from oral single co-administration of PTX and compound **5c**.

### 2.8. Data Analysis

Statistical analysis was conducted using Tukey and Dunnett T3′s tests in conjunction with a one-way analysis of variance (ANOVA) for accumulation, efflux, and PK studies, respectively. Free GraphPad Prism was used for statistical analyses (Version 8.00, La Jolla, San Diego, CA, USA). Mean data were presented with standard deviations (SD). A *p* value < 0.05 was considered statistically significant.

## 3. Results

### 3.1. Synthesis of Ferulic Amides, FA Derivatives

FA derivatives were synthesized as shown in Scheme 1. Ferulic acid (1) was protected with methyl alcohol to give methyl ester (2). The ester was then reacted with potassium carbonate and different alkyl halides such as propyl iodide, isobutyl iodide or benzyl bromide to afford corresponding esters (3) and purified by column chromatography. Then, the protected methyl ester was hydrolyzed with 2 N sodium hydroxide to give the acids (4). The coupling of acid group with various amines using PyBOP as coupling agent resulted in the desired amides (5).

### 3.2. P-gp Inhibitory Effect of FA Derivatives in Vitro

VER (a positive control) and FA derivatives were shown to be not toxic to MCF-7/ADR cells at a concentration of 100 µM (Appendix A). When eight FA derivatives (Figure 1, 100 µM) were treated with DNM in MCF-7/ADR cells, compounds **5c**, **5f**, **5g** and **5h** decreased the IC_50_ values of DNM to about 1/5 of negative control (DNM alone) value (Table 1). Even compounds **5c** and **5h** dramatically reduced the IC_50_ values of DNM (2.2 and 2.6 µM, respectively) which was lower than that of VER-treated group (3.1 µM).

Based on this result, compounds **5c**, **5f**, **5g** and **5h** were selected and tested for their effects on cellular accumulation and efflux of tritiated DNM (Figure 2). Four compounds and VER exhibited significant effects on both tests compared to the control (*P* < 0.001 and 0.0001). Among them, compounds **5c**, **5e** and **5h** enhanced the accumulation of [^3^H]-DNM (438.4 ± 4.9%, 382.1 ± 20.4% and 426.1 ± 26.2%, respectively) more than VER (335.1 ± 14.7%) (Figure 2A). Coherently, the efflux of [^3^H]-DNM in the presence of compounds **5c**, **5e** and **5h** (43.1 ± 4.6%, 39.9 ± 3.8% and 43.3 ± 0.9%, respectively) was substantially reduced compared to the control or VER (63.0 ± 3.3% or 48.0 ± 1.1%, respectively) (Figure 2B).

Furthermore, compared to VER (2.24), a well-known ATPase stimulator, four FA derivatives exhibited higher values (4.02, 3.35, 5.19 and 2.79) for ATPase activity at the concentration of 100 µM. Among them, compound **5c** stably increased ATPase activity in a concentration-dependent manner (Table 2). Based on this result, four FA derivatives may inhibit P-gp activity by depleting ATP as an ATPase stimulator.

### 3.3. BA Enhancing Effect of Compound **5c** in Vivo

The in vitro results suggested that compound **5c** was the most potent P-gp inhibitor. As shown in Table 3, co-administration of the lowest dose of compound **5c** (0.5 mg/kg) did not alter any PK parameters of PTX, as compared to PO control (PTX alone). However, with higher doses of compound **5c** (2 and 5 mg/kg), the AUC_inf_ of PTX was increased (control; 773 ± 169 ng·h/mL vs. 5 mg/kg; 1456 ± 367 ng·h/mL, *p* < 0.05) and Cl/F of PTX was decreased approximately to one half (control; 8.22 ± 1.84 L/h vs. 2 mg/kg; 4.10 ± 1.53 L/h (*p* < 0.05) and 5 mg/kg; 4.53 ± 1.04 L/h (*p* < 0.05), respectively). The t_1/2_ was extended over two-fold (control; 2.8 ± 0.86 h vs. 5 mg/kg; 6.8± 1.3 h, *p* < 0.01) (Table 3). The drug exposure in the body was enhanced due to the reduction of oral clearance when it was co-administered with compound **5c**, except for a dose of 0.5 mg/kg (Figure 3). On the other hand, the C_max_ of PTX and Vz/F were not significantly changed in the presence of the compound.

## 4. Discussion

FA enhanced the cytotoxicity of PTX and doxorubicin in human nasopharyngeal cancer cells by inhibiting the P-gp function [15,16]. FA enhanced cellular accumulation of calcein-acetoxymethyl ester and rhodamine123, well-known P-gp substrates, in P-gp overexpressed cells in a dose-dependent manner [15]. In silico study, FA showed hydrogen bonding and hydrophobic interaction with transmembrane domain of P-gp [15]. In our study, newly synthesized FA derivatives including compound **5c** increased the cytotoxicity and cellular accumulation of DNM in P-gp overexpressed breast cancer cells. Moreover, some parts of structure in the FA and its derivatives showed a similarity with VER, a well-known P-gp inhibitor. Because of the similar parts of the structure, FA derivatives might show the effect on the reversal of the P-gp-mediated MDR in cancer cells.

In addition, the dose-dependent decrease in ATPase activity of VER was also observed in our previous study [10]. However, all the values were over 2 (as a ratio to blank), suggesting ATPase stimulating effect of VER. In our previous studies, the lower dose (50 μΜ) of VER showed less effects on DNM accumulation and efflux [8] than the higher dose (100 μΜ) of VER [10,19]. Therefore, we used 100 μΜ VER and FA derivatives to observe P-gp inhibitory activity as much as possible in in vitro studies. According to the results of ATPase activity assay, four FA derivatives may inhibit P-gp activity by depleting ATP as an ATPase stimulator because a ratio to blank was over 2.

FA derivative, compound **5c** chosen as the most potent P-gp inhibitor among the eight synthesized FA derivatives from in vitro studies increased PTX BA by reducing oral PTX clearance in vivo. While C_max_ of PTX was not significantly altered in the presence of compound **5c**, the t_1/2_ of PTX was significantly extended when PTX was co-administered with 2 or 5 mg/kg of the FA derivative (Table 3). The increased AUC_inf_ of the drug observed in the higher dose groups (2 and 5 mg/kg) was a result of the decreased PTX elimination. In addition, the reduced variation was monitored in the plasma PTX concentration-time profiles of the higher dose groups (2 and 5 mg/kg), as compared with control or low dose group (0.5 mg/kg).

Previously we reported other novel P-gp inhibitors such as coumarin derivative [8], phenylbutenoid dimer [9] and xanthone analogue [10]. They and VER [8] enhanced BA (9.6-fold, 1.8-fold and 2.5-fold and 3.0-fold at 5 mg/kg, respectively) and C_max_ of PTX when they were co-administered with P-gp substrate anticancer drug, PTX. It means that these inhibitors worked at drug absorption phase by modulating the intestinal P-gp function. On the other hand, the FA derivative in the present study appeared to modulate P-gp function in the elimination phase, unlike other P-gp inhibitors reported previously [8,9,10].

Our study showed the potential P-gp modulating effect of FA derivative, compound **5c,** on PTX disposition. Indeed, P-gp efflux transporters are expressed in the hepatic canalicular membrane for biliary excretion and in the proximal tubule of the kidney for renal excretion [2,4]. In addition, PTX has been reported to be eliminated by hepatic metabolism by mainly CYP2C8 [21], renal excretion [22] and biliary excretion [23]. It has been reported that FA is quickly absorbed in the intestine within 5–15 min and 30 min in rats and humans, respectively [24], and is mainly excreted through kidney [25]. Therefore, the interaction between FA derivative, absorbed at GI tract, and PTX on P-gp efflux transporters might occur during the elimination process. There is a lack of knowledge about the effect of FA derivatives and/or their metabolites on the hepatic metabolism and biliary excretion, and it needs to be clarified.

According to the reports about the relationship between FA and renal function in rodents, FA gives the positive effect on the kidney injury [26,27,28]. In lipopolysaccharide (LPS)-induced acute kidney injured mice, the phenolic compound (50 mg/kg and 100 mg/kg, intraperitoneal (IP) injection, 1 h before and 2 h after a single IP injection of LPS) reduces apoptosis and inflammation but increases adenosine generation [27], and in diabetes-induced rats, FA (50 mg/kg, orally for 8-week) protects hyperglycemia-induced kidney damage by antioxidation, anti-inflammation and autophagy [28]. In the present study, a single dose of FA derivative (0.5, 2 or 5 mg/kg) may not affect renal function in healthy rats to reduce PTX elimination. However, further studies are required to examine the effect of FA derivatives on renal function and their toxicities in the body following single or multiple dosing.

## 5. Conclusions

Four FA derivatives exhibited P-gp inhibitory function among eight synthesized compounds, and compound **5c** was selected as the most potent P-gp inhibitor candidate in in vitro investigation. This compound significantly improved the BA of PTX by approximately twofold at the dose of 5 mg/kg, when it was co-administered with PTX in rats. Therefore, the co-administration of compound **5c** could offer a therapeutic benefit in oral administration of P-gp substrate drugs by reducing drug elimination. Further studies are required to examine the effect of compound **5c** on PTX metabolism to clarify the mechanism for the decreased PTX elimination caused by the FA derivative and to investigate the effect of the FA derivative, compound **5c** on renal function.

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
