# Peer review of "Pharmacokinetic Alteration of Paclitaxel by Ferulic Acid Derivative"

_pharmaceutics, 2019, doi:10.3390/pharmaceutics11110593_

Round 1

Reviewer 1 Report

The paper is understandable but some experiments are missing.

Please find below some comments for consideration:

Authors did not observe dose-relationship in vivo. The dose of 2 mg/kg of compound 3 showed more effects than 5 mg/kg. Authors should propose hypothesis why, e.g., auto inhibition.... PK of PTX are characterized by a very limited renal clearance. The first part of the discussion seems out of scope, authors may improve that. Why was VER not administered in vivo, it would have been nice to have a positive control in vivo. Why was PTX only administered p.o. An i.v. formulation is available and authors would have been able to discriminate the effect at the intestine level. The title is incorrect, only one DA derivative was investigated. Please check some edits/wording (L22: level of AUC, L284: three....)

Author Response

Thank you for your comments and suggestions on our manuscript. We have taken your critiques into account for revising the manuscript. It has been led to strengthen the manuscript overall. Below we provided the point-to-point responses to your comments. In the revised manuscript, our responses are marked as yellow highlights.

Point 1: Authors did not observe dose-relationship in vivo. The dose of 2 mg/kg of compound 3 showed more effects than 5 mg/kg.

Response 1: I agree with your opinion. In our study, 2 mg/kg of compound 3(5c) was more effective than 5 mg/kg of the compound in terms of pharmacokinetic parameters of PTX. Therefore, 2 mg/kg of compound 3(5c) might be optimal dose for increasing relative bioavailability of PTX. However, because 2 mg/kg of compound 3(5c) did not show statistical significance on the AUCinf of PTX versus control (PTX only) whereas 5 mg/kg of the compound significantly increased the AUCinf (Table 3), we did not claim that 2 mg/kg of compound 3(5c) showed more effects than 5 mg/kg. Rather, we mentioned the changes in PK parameters of PTX at higher doses (2 and 5 mg/kg) as compared with those of 0.5 mg/kg dose (yellow highlight on the first paragraph in 3.3.)’.

Point 2:  Authors should propose hypothesis why, e.g., auto inhibition.... PK of PTX are characterized by a very limited renal clearance.

Response 2: Thank you for your comments. In the beginning, we expected that compound 3(5c) increased Cmax of PTX by inhibiting P-gp expressed in the gastrointestinal tract at the absorption phase. However, we did not see any significant change in Cmax of PTX. Instead, we found significant decrease in total clearance of PTX along with significant increase in t1/2 of PTX. Because PTX is eliminated by hepatic metabolism, biliary excretion and renal excretion and P-gp is located at the hepatic canalicular membrane and renal proximal tubule, we suggested that compound 3(5c) might inhibit P-gp located at both hepatic canalicular membrane and renal proximal tubule, resulting in reduction of renal and/or biliary excretion of PTX in the Discussion.

Point 3:  The first part of the discussion seems out of scope, authors may improve that.

Response 3: As followed by reviewer’s comments, we checked and improved the first part of the Discussion (yellow highlight).

Point 4: Why was VER not administered in vivo, it would have been nice to have a positive control in vivo.

Response 4: Thank you for your comments. In our previous studies, we found that VER increased the BA of PTX (Eur J Pharmacol 2014:723:381; Biomol Ther 2017:25:553). The result of co-administration of PTX and VER was added in the third paragraph of Discussion (yellow highlight).

Point 5: Why was PTX only administered p.o. An i.v. formulation is available and authors would have been able to discriminate the effect at the intestine level.

Response 5: Thank you for your comments. The absolute bioavailability of PTX is very low (4 - 7%) due to its poor water solubility and low permeability (Int J Pharm 1998:172:1). Because oral route is favorable for patients, we tried to develop oral delivery formulation of PTX with P-gp inhibitor which can increase intestinal permeability. In the beginning, we expected that compound 3(5c) increased Cmax of PTX by inhibiting P-gp expressed in the gastrointestinal tract at the absorption phase as reported previously (J Nat Prod 2013:76:2277; Eur J Pharmacol 2014:723:381; Eur J Med Chem 2015:93:237). However, we did not see any significant change in Cmax of PTX. Therefore, we concluded that compound 3(5c) increased BA of PTX by inhibiting P-gp located in the eliminating organs after its fast absorption. Here we reported what we found in this study as a short communication. Further studies are required to examine the effect of compound 3(5c) on PTX pharmacokinetics after IV injection.

Point 6: The title is incorrect, only one DA derivative was investigated.

Response 6: As followed by reviewer’s comments, we changed the title to “Pharmacokinetic Alteration of Paclitaxel by Ferulic Acid Derivative”.

Point 7: Please check some edits/wording (L22: level of AUC, L284: three....)

Response 7: As followed by reviewer’s comments, the corrections were yellow highlighted.

Thank you for your consideration of this work.

Sincerely,

Hwa Jeong Lee, Ph.D.

Professor

Reviewer 2 Report

The authors have performed considerable amount of experimental work including synthesizing several DA derivatives in the study to evaluate their function on P-gp in vitro and in vivo systems.

As compared to authors’ previous work studying effect of coumarin derivative-mediated inhibition of P-glycoprotein on oral bioavailability and therapeutic efficacy of paclitaxel where improved the relative bioavailability (RB) of PTX to 961% was observed, the enhancement of oral bioavailability using Dopamine Derivatives is less pronounced in this current study. The observed AUCinf 2-fold versus the control was largely due to terminal concentration difference. Adding details of in vivo portion of the study including study design (e.g., # of animals used per timepoint), a summary of the bioanalytical method for plasma concentration measurement (including lower limit of quantitation), and discussion on animal variation and effects will help strengthen the conclusion of in-vivo data. Paclitaxel is predominantly by the liver by cytochromes P450 2C8 and 3A4. There is no evaluation on whether Dopamine Derivatives and/or their metabolite played an inhibitory role of hepatic enzymes in the observed relative bioavailability increase. Such potential impact needs to be addressed in the discussion. Line 199-200: Clarify if AUC is AUCinf Line 284: “Three are several…” should be “There are several…”

Author Response

Thank you for your comments on our manuscript. We have taken your critiques into account for revising the manuscript. It has been led to strengthen the manuscript overall. Below we provided the point-to-point responses to your comments. In the revised manuscript, our responses are marked as light green highlights.

Point 1: Adding details of in vivo portion of the study including study design (e.g., # of animals used per time point), a summary of the bioanalytical method for plasma concentration measurement (including lower limit of quantitation), and discussion on animal variation and effects will help strengthen the conclusion of in-vivo data.

Response 1: Thank you for your comments. As followed by your comments, we added or modified the part of in vivo study in Methods (the end of third paragraph in 2.6. PK study) and in Discussion (the second paragraph). All the sentences were light green highlighted.

Point 2:  Paclitaxel is predominantly by the liver by cytochromes P450 2C8 and 3A4. There is no evaluation on whether Dopamine Derivatives and/or their metabolite played an inhibitory role of hepatic enzymes in the observed relative bioavailability increase. Such potential impact needs to be addressed in the discussion.

Response 2: I agree with your opinion. Paclitaxel has very low bioavailability due to its poor water solubility and low permeability. For the convenience of cancer treatment, we tried to develop oral formulation of PTX with P-gp inhibitor which can increase intestinal permeability of PTX. However, compound 3(5c) did not enhance absorption of PTX. Instead, this compound increased BA of PTX by reduction in total clearance of PTX. This means that compound 3(5c) can absorb very quickly in the intestine without significant inhibition on P-gp existed in the gastrointestinal tract. Then, it can interact with P-gp located in hepatic canalicular membrane and/or renal proximal tubule in order to reduce PTX elimination. Here, we reported what we found in this study as a short communication. Because we also consider the possible effect of compound 3(5c) on PTX metabolism, it was mentioned in Discussion and Conclusions (light green highlight) and will be studied further.

Point 3:  Line 199-200: Clarify if AUC is AUCinf

Response 3: As followed by your comments, we changed ‘AUC’ to ‘AUCinf’ (light green highlight).

Point 4:  Line 284: “Three are several…” should be “There are several…”

Response 4: The part was deleted.

Thank you for your consideration of this work.

Sincerely,

Hwa Jeong Lee, Ph.D., Professor

Reviewer 3 Report

The manuscript has major issues and most of them are presented here. Is they are not resolved, I would recommend the rejection of this paper!

Row 34: the affirmation “largely responsible for the failure of cancer chemotherapy” is not sustained by data. In my view, is also false. In many cases, like EGFR inhibitors, target mutations are the main cause of resistance. Add some examples of drugs to back up this, but still I think is excessive.

Row 39: “because of the biological properties of DA, the catecholamine structure has been used as a core for numerous drugs”. I don’t agree with this statement. First, because catecholamines scaffold is also present in adrenaline and noradrenaline and many drugs were design to mimic those, and not only dopamine. The second problem is that the most used part is the β-phenylethylamine, and not catecholamine because is a substrate for COMT and is very quickly metabolized. Even the compounds synthesized by the authors are not exactly cathecolamine! The phenol groups are blocked ad the nitrogen is not linked by an ethyl fragment.

Row 46: Why the scheme is presented in the introduction?

Row 46-48: The scheme is not correct! The compounds should be numbered. R is methyl, and not ethyl! In the case of R1-X, X seems to be iodide, and bromide, not chloride!

Row 54: explain all abbreviation in the text. Like PTX!

Row 59: what does c.H2SO4 mean?

Row 65: “To the methyl ferulate 0.5 g (2.4 mmol) in dry acetone (6 mL) solution was added K2CO3”. Be careful to the language! It sounds strange! There are other example in the whole manuscript.

Row 66: what was the source for iodopropane? See row 74: 1-iodo-2-methylpropane. See row 80: benzyl bromide. Row 107: i-Pr2NEt

In my view the compounds are inspired by verapamil, and not dopamine. Just compare the structure of 6 with verapamil! I advise the authors to change their title and all the introduction to be better suited.

The title should be changed because the compounds are not exactly dopamine derivatives and also the pharmacokinetics modification are not that relevant and were tested only on one compound, not on ALL derivatives as suggested!

Row 68: hexane

Row 69 and elsewhere in the paper. How was the mp measure?

Row 73: “The same procedure as 2.2.1”. It should be 2.2.2.1. The same in row 79

Row 105 (and scheme 1): what is PyBOB and what was the source for it?

Ver abbreviation should be presented in the text

The numbering of the compound should be propoxy derivatives 1 and 5, and isobutoxy as 2 and 6.

The major problem is that the compound are not really original. See pubchem for compound 1: CID 1631920, compound 2: CID 71747693, compound 3: CID 19173937, compound 4: CID 71747160, compound 5: CID 71747691, compound 6: CID 71747332, compound 7: CID 71747333, compound 8: CID 1747331.

I don’t understand the presentation of all synthesis methods and NMR studies, when the compounds are already published. The authors have to do a little effort to improve their research with relevant data from literature. Some references should be added about these compounds.

Row 139: N-Benzylamine???

Row 160: how were the IC50 values calculated? The table 1 present what I presume an average values. But how confident is this value? Some statistical parameters should be added!

Row 167. A letter is shown in red! The section 211-214 is in bold. Why?

Row 177: this study was approved in 2012???

Row 211: “propyl iodide, isobutyl bromide, or benzyl iodide”. Incorrect! The authors should have checked the manuscript before submission!

Row 216: Figure 1, compound 5 is missing the double bond

Row 218: “VER (a positive control) and DA derivatives were shown to be not toxic to MCF-7/ADR cells at a concentration of 100 µM”. How was this conclusion obtained? Based on what data?

Row 220: “compounds 3, 6, 7 and 8 decreased the IC50 values of DNM (Table 1)”. This can indicate a synergic cytotoxic effect, and not the inhibition of P-gp. Scientifically speaking, the inhibition of P-gp is probabily the mechanism, but the authors have no proof of that. Please correct this problem! Take also in consideration the confidence range of the IC50 values!!!

How can they explain the difference between compounds 1 and 2, considering the very similar structures?

Row 248: I don’t understand why in the case of verapamil the effect on ATPase is decreasing with concentration. Maybe the 100 μM dose is to high? Maybe some smaller doses should be used? The same is observed for compound 8. The authors seem to ignore this interesting phenomen. Why? Also, the same problem. No statistical data for the results!

Row 283: Or it could mean than the compound 3 is an inhibitor of CYP2C8! The authors should be careful with their conclusions!!!

Row 284-289. This section is not related to the paper. The authors are to demonstrate that the new compounds have dopamingergic effects. Until they do, this are just speculations and hypotheses.

Author Response

Thank you for your comments on our manuscript. We have taken your critiques into account for revising the manuscript. It has been led to strengthen the manuscript overall. Below we provided the point-to-point responses to your comments. In the revised manuscript, our responses are marked as sky blue highlights.

Point 1: Row 34: the affirmation “largely responsible for the failure of cancer chemotherapy” is not sustained by data. In my view, is also false. In many cases, like EGFR inhibitors, target mutations are the main cause of resistance. Add some examples of drugs to back up this, but still I think is excessive.

Response 1: Thank you for your comments. However, it has been reported that P-gp among ABC transporters is closely associated with MDR (Nat Struct Mol Biol 2016:23:487). As followed by your comments, the sentence was corrected as “is one of the responsible reasons for the failures of cancer chemotherapy” and the following references were added: Nanomedicine 2010:5:597; Biomed Pharmacother 2019:118:109233; Biochim Biophys Acta Gen Subj 2019:1863:1390.

Point 2:  Row 39: “because of the biological properties of DA, the catecholamine structure has been used as a core for numerous drugs”. I don’t agree with this statement. First, because catecholamines scaffold is also present in adrenaline and noradrenaline and many drugs were design to mimic those, and not only dopamine. The second problem is that the most used part is the β-phenylethylamine, and not catecholamine because is a substrate for COMT and is very quickly metabolized. Even the compounds synthesized by the authors are not exactly cathecolamine! The phenol groups are blocked ad the nitrogen is not linked by an ethyl fragment.

Response 2: I agree with you opinion. As followed by your comments, we changed ‘DA’ to ‘ferulic acid (FA)’ in the text.

Point 3:  Row 46: Why the scheme is presented in the introduction?

Response 3: As followed by your comments, we replaced the scheme to ‘Results’

Point 4: Row 46-48: The scheme is not correct! The compounds should be numbered. R is methyl, and not ethyl! In the case of R1-X, X seems to be iodide, and bromide, not chloride!

Response 4: As followed by your comments, the scheme was changed and the compounds were numbered.

Point 5: Row 54: explain all abbreviation in the text. Like PTX!

Response 5: As followed by your comments, the abbreviation was explained as ‘ATP-binding cassette (ABC)’ in line 29-30 and ‘Paclitaxel (PTX)’ in line 53.

Point 6: Row 59: what does c.H2SO4 mean?

Response 6: It was corrected as ‘conc. sulfuric acid’.

Point 7: Row 65: “To the methyl ferulate 0.5 g (2.4 mmol) in dry acetone (6 mL) solution was added K2CO3”. Be careful to the language! It sounds strange! There are other example in the whole manuscript.

Response 7: We corrected as ‘To the methyl ferulate 0.5 g (2.4 mmol) in dry acetone (6 mL) solution, potassium carbonate (K2CO3) 0.4 g (2.9 mmol) was added’. In addition, we arranged the synthetic part in ‘Methods’.

Point 8:  Row 66: what was the source for iodopropane? See row 74: 1-iodo-2-methylpropane. See row 80: benzyl bromide. Row 107: i-Pr2NEt

Response 8: We purchased all reagents as commercial grade. In ‘Materials’, it was mentioned that ‘Ferulic acid, sulforhodamine B (ARB), (+/-) verapamil hydrochloride and other reagents for synthesis were purchased from Sigma-Aldrich (St. Louis, USA)’ and ‘All reagents and solvents were supplied as synthetic grades’.

Point 9: In my view the compounds are inspired by verapamil, and not dopamine. Just compare the structure of 6 with verapamil! I advise the authors to change their title and all the introduction to be better suited.

Response 9: No, the compounds were not inspired by VER. In this study, the compounds were originated from FA. Initially, these FA derivatives was developed as neuro-protection compound for targeting to dopamine neuron, and those had been called as DA derivatives. Now, we corrected their names as FA derivatives. However, some part of structure showed a similarity between FA derivatives and VER, as you commented. Because of the similar part of structure, FA derivatives might show the effect on the reversal of the P-gp-mediated MDR in cancer cells. The enhancement of cytotoxicity and accumulation of anticancer drug, DNM in P-gp overexpressed breast cancer cells was also observed in our studies. Moreover, in silico study, FA showed hydrogen bonding and hydrophobic interaction with transmembrane domain of P-gp (Eur J Pharmacol 2016:786:194) like VER.

Title, Introduction and Discussion were changed to be suitable for FA or/and FA derivatives.

Point 10: The title should be changed because the compounds are not exactly dopamine derivatives and also the pharmacokinetics modification are not that relevant and were tested only on one compound, not on ALL derivatives as suggested!

Response 10: As followed by your comments, we changed the title. Among eight FA derivatives, compound 5c ((E)-3-(4-(benyloxy-3-methoxyphenyl)-1-(piperidin-1-yl)prop-2-en-1-one) was chosen to examine the effect on PTX pharmacokinetics because it was most effective to inhibit P-gp function in vitro.

Point 11: Row 68: hexane

Response 11: It was corrected as your comment.

Point 12: Row 69 and elsewhere in the paper. How was the mp measure?

Response 12: Melting points were measured on an electro thermal digital melting point (Buchi, Germany) without calibration. We added this explanation in Materials and Methods.

Point 13:  Row 73: “The same procedure as 2.2.1”. It should be 2.2.2.1. The same in row 79

Response 13: The experimental part was deleted.

Point 14: Row 105 (and scheme 1): what is PyBOB and what was the source for it?

Response 14: It was corrected as ‘PyBOP (benzotriazol-1-yl-oxytripyrrolidinophosphonium hexafluorophosphate)’. Its source was commercial grade reagent purchased from Sigma-Aldrich (St. Louis, USA), as we mentioned in Materials.

Point 15: Ver abbreviation should be presented in the text

Response 15: We already mentioned ‘VER’ as the abbreviation of ‘Verapamil’ in the text of ‘2.3. Methods’.

Point 16: The numbering of the compound should be propoxy derivatives 1 and 5, and isobutoxy as 2 and 6.

Response 16: The compounds were numbered from 5a to 5h. The compounds were mentioned as FA derivatives in the text.

Point 17: The major problem is that the compound are not really original. See pubchem for compound 1: CID 1631920, compound 2: CID 71747693, compound 3: CID 19173937, compound 4: CID 71747160, compound 5: CID 71747691, compound 6: CID 71747332, compound 7: CID 71747333, compound 8: CID 1747331.

Response 17: Except for compound 3(5c), all the compounds were not published, yet. All the synthesized compounds including 3(5c) have been registered as a patent (KR10-2013-0099676) by Prof. Hea-Young Park and colleagues in our group since 2013.

Point 18: I don’t understand the presentation of all synthesis methods and NMR studies, when the compounds are already published. The authors have to do a little effort to improve their research with relevant data from literature. Some references should be added about these compounds.

Response 18: Because we do not publish the synthesis parts of the FA derivatives, there are no references about the parts including NMR studies, yet. It can be found through ‘SciFinder’ program. However, the compound 3(5c) published by another group (Bioorg Med Chem 2018:26:5672, i.e. compound 40 in ref.) is introduced as referring the literature.

Point 19: Row 139: N-Benzylamine???

Response 19: It was corrected to Benzylamine.

Point 20: Row 160: how were the IC50 values calculated? The table 1 present what I presume an average values. But how confident is this value? Some statistical parameters should be added!

Response 20: For the cytotoxicity assay, cells were seeded in 96-well plates at 5,000 cells per well and incubated for 24 h. DNM was added to final concentrations of 1.0 x 10-7 to 1.0 x 10-4 M with or without each FA derivative or VER (positive control) at a concentration of 100 uM. After a 2-h incubation, cells were washed and given 200 μL of fresh culture media followed by incubation for 72 h. DNM cytotoxicity was determined using a modified SRB staining assay. IC50 values of DNM in the presence or absence of FA derivatives were calculated with Table Curve 2D® version 5.01 software (Systat Software Inc., San Jose, CA, USA). (Eur J Med Chem 2015:93:237).

Because cytotoxicity assay is a screening process to find possible P-gp inhibitor candidates, reporting mean values seems to be suitable in this study. After the possible P-gp inhibitor candidates were selected from the cytotoxicity assay, the selected compounds were further studied for their effects on cellular accumulation and efflux of DNM. The cellular accumulation and efflux data were reported with mean +/- SD and statistical analysis results.

Point 21: Row 167. A letter is shown in red! The section 211-214 is in bold. Why?

Response 21: Both parts pointed were corrected (Please see sky blue highlights).

Point 22: Row 177: this study was approved in 2012???

Response 22: Yes, the animal study protocol was approved in 2012 and has been renewed since 2012.

Point 23: Row 211: “propyl iodide, isobutyl bromide, or benzyl iodide”. Incorrect!

Response 23: Thank you for your comments. We corrected as ‘propyl iodide, isobutyl iodide, or benzyl bromide’.

Point 24: Row 216: Figure 1, compound 5 is missing the double bond

Response 24: It was corrected as followed by your comments.

Point 25: Row 218: “VER (a positive control) and DA derivatives were shown to be not toxic to MCF-7/ADR cells at a concentration of 100 µM”. How was this conclusion obtained? Based on what data?

Response 25: In our laboratory, we are searching for P-gp inhibitors to use in combination with P-gp substrate anticancer drug orally. In the beginning, P-gp inhibitor candidates are selected from various series compounds by simple cytotoxicity assay as a screening process. The compounds 1-8 (5a-5h) were not significantly cytotoxic up to 100 μΜ. This preliminary data was added as the supplementary Table 1. In case of VER, the concentration of 100 μΜ has been confirmed not to cytotoxic in previous reports (Eur J Med Chem, 2015:93:237).

Point 26: Row 220: “compounds 3, 6, 7 and 8 decreased the IC50 values of DNM (Table 1)”. This can indicate a synergic cytotoxic effect, and not the inhibition of P-gp. Scientifically speaking, the inhibition of P-gp is probabily the mechanism, but the authors have no proof of that. Please correct this problem! Take also in consideration the confidence range of the IC50 values!!!

Response 26: Thank you for your comments. We did not mention directly that the results of cytotoxicity assay indicated the P-gp inhibition in the text. Cytotoxicity assay is a screening process to find possible P-gp inhibitor candidates, though increased cytotoxicity of DNM by the compounds can come from different mechanisms (i.e. synergic cytotoxic effect). Therefore, P-gp inhibitory effect of FA derivatives was confirmed by DNM accumulation and efflux studies using the P-gp over-expressing cells. In the present study, four compounds (5c, 5f, 5g and 5h) were selected as possible P-gp inhibitor candidates from the cytotoxicity results because their IC50 values were similar to VER (a well-known P-gp inhibitor) or only about 2-fold higher than the value of VER. Because the IC50 values of the rest compounds (5a, 5b, 5d and 5e) were more than 10 or higher, they were not chosen for further studies. And then, the selected four compounds were examined for their P-gp inhibitory activities through the cellular accumulation and efflux tests (J Nat Prod 2013:76:2277; Eur J Med Chem, 2015:93:237; J Pharm Pharmacol 2018:70:234).

Point 27: How can they explain the difference between compounds 1 and 2, considering the very similar structures?

Response 27: Sometimes, one methyl difference in structure causes different efficacy. In this study, the effect of both compounds on DNM cytotoxicity was different at some degree. However, because the effect of compound 5b(2) on DNM cytotoxicity was not strong enough as compared with compounds 5c, 5f, 5g and 5h, it was not further studied for cellular accumulation, efflux and ATPase activity.

Point 28: Row 248: I don’t understand why in the case of verapamil the effect on ATPase is decreasing with concentration. Maybe the 100 μM dose is to high? Maybe some smaller doses should be used? The same is observed for compound 8. The authors seem to ignore this interesting phenomen. Why? Also, the same problem. No statistical data for the results!

Response 28: Thank you for the comments. The dose-dependent decrease in ATPase activity of VER was also observed in our previous study (Eur J Med Chem, 2015:93:237). However, all the values were over 2 (as a ratio of blank), suggesting ATPase stimulating effect of VER. In our previous studies, the lower dose (50 μΜ) of VER showed less DNM accumulation and efflux results (Eur J Pharmacol 2014:723:381) than higher dose (100 μΜ) of VER (Eur J Med Chem, 2015:93:237; J Pharm Pharmacol 2018:70:234). Therefore, we used 100 μΜ VER and FA derivatives to observe P-gp inhibitory activity as much as possible in in vitro studies. According to the results of ATPase activity assay, four FA derivatives may inhibit P-gp activity by depleting ATP as an ATPase stimulator because a ratio of blank was over 2.

Point 29: Row 283: Or it could mean than the compound 3 is an inhibitor of CYP2C8! The authors should be careful with their conclusions!!!

Response 29: We mentioned that ‘there is a lack of knowledge about the effect of FA derivative and/or their metabolites on elimination process including the hepatic metabolism and biliary excretion, and it needs to be clarified’ at the end of the fourth paragraph of ‘Discussion’. Therefore, Further studies are required to examine the effect of compound 5c(3) on PTX metabolism to clarify the mechanism for the decreased PTX elimination caused by the FA derivative.

Point 30: Row 284-289. This section is not related to the paper. The authors are to demonstrate that the new compounds have dopamingergic effects. Until they do, this are just speculations and hypotheses.

Response 30: As followed by your comments, the section was deleted.

Thank you for your consideration of this work.

Sincerely,

Hwa Jeong Lee, Ph.D.

Professor

Reviewer 4 Report

The reviewer does not support the acceptance for this manuscript.

(1) The safety data of such p-Gp inhibitors is missing. 

(2) The animal handling procedures are not in details.

(3) HPLC-UV is not sensitive enough for pharmacokinetic profiling. 

(4) Most importantly, based on Figure 3, the reviewer concludes that the tested compound actually do not possess strong in vivo efficacy. The difference in parameters is mainly due to the difference observed in the samples collected at 24 h.

Author Response

Thank you for your comments on our manuscript. We have taken your critiques into account for revising the manuscript. It has been led to strengthen the manuscript overall. Below we provided the point-to-point responses to your comments.

Point 1: The safety data of such p-Gp inhibitors is missing. 

Response 1: The compounds 1 ~ 8(5a ~ 5h) were not cytotoxic up to 100 μΜ in vitro. We added the data as the supplementary table 1. In in vivo study, the compound 3(5c) was not toxic up to 5 mg/kg in the presence of anticancer drug, PTX (25 mg/kg) following oral administration. In our laboratory, we are searching for P-gp inhibitors to use in combination with P-gp substrate anticancer drug orally. So far, maximum dose of P-gp inhibitors used orally in animal study was 10 mg/kg (Eur J Pharmacol 2014:723:381). In a complex of anticancer drug (main drug) and P-gp inhibitor (absorption enhancer or chemo-sensitizer), dose of P-gp inhibitor should be as low as possible if it can show absorption enhancing effect or chemo-sensitizing effect of anticancer drug. In our study, 2 mg/kg of compound 3(5c) was more effective than 5 mg/kg of the compound in terms of pharmacokinetic parameters of PTX. Therefore, 2 mg/kg of compound 3(5c) might be optimal dose for increasing relative bioavailability of paclitaxel. Consequently, the doses of compound 3(5c) used in this study (up to 5 mg/kg) were safe for the animal study. However, further study is required to examine the toxicity of the compound 3(5c).

Point 2:  The animal handling procedures are not in details.

Response 2: According to ‘Instructions for Authors’ (https://www.mdpi.com/journal/pharmaceutics/instructions), Materials and Methods: They should be described with sufficient detail to allow others to replicate and build on published results. New methods and protocols should be described in detail while well-established methods can be briefly described and appropriately cited. Give the name and version of any software used and make clear whether computer code used is available. Include any pre-registration codes.

Because the animal handling procedures were already set up in our laboratory and described in detail reported previously (J Nat Prod 2013:76:2277, Eur J Med Chem, 2015:93:237, J Pharm Pharmacol 2018:70:234), we briefly described about the animal handling procedures and appropriately cited in ‘Materials and Methods’, as followed by ‘Instructions for Authors’.

Point 3:  HPLC-UV is not sensitive enough for pharmacokinetic profiling. 

Response 3: Although LC-MS/MS method is recently used for PK profiling due to its sensitivity and selectivity (Talanta 2013:116:2277), HPLC-UV method is still widely and preferably used in many laboratories because LC-MS/MS instrumentation is much expansive and requires highly qualified operator than HPLC-UV analysis (Clin L B Med 2016:36:635). Moreover, LC-MS/MS technique can bring the problem that ion suppression caused by sample matrix brings significantly different quantification values from one sample (Talanta 2013:115:104). Rather this problem combined with the strong point, the high sensitivity, sometimes, brings a maximized error.

In the present study, the simple HPLC-UV method validated by FDA guideline for the quantification of PTX in rat plasma showed LLOQ at 0.01 µg/mL, linearity over the concentration range between 0.01 and 10 µg/mL (R2 = 0.9999), and accepted precision & accuracy (2.4-9.1% & 92.1-102% for intra- and inter-day, respectively) (written at the end of the third paragraph of 2.6. PK study in Methods). Although this HPLC-UV method is not sensitive as much as LC-MS/MS analytical methods, it is suitable for the determination of PTX in rat plasma for PK study.

Point 4: Most importantly, based on Figure 3, the reviewer concludes that the tested compound actually do not possess strong in vivo efficacy. The difference in parameters is mainly due to the difference observed in the samples collected at 24 h.

Response 4: PTX was not detected in rat plasma sample collected at 24 h in both PTX control and low dose (0.5 mg/kg of compound 3(5c)) groups, while it was measured at 24 h in higher dose groups (2 and 5 mg/kg of the compound). Because this phenomenon was observed in all animals studied (n = 4~5 per each group), we could not ignore the plasma drug concentrations determined at 24 h. As you mentioned, the differences in PK parameters observed in different doses of the compound was mainly resulted from the determination of PTX in rat plasma at the last time point, 24 h. However, plasma drug concentrations measured up to 10 h of higher dose group were also greater than those of both PTX control and low dose group. The higher doses of compound 3(5c) significantly increased AUCinf and t1/2 with significant decrease in total clearance. Therefore, we suggested that compound 3(5c) might affect elimination of PTX by inhibiting P-gp located in hepatic canalicular membrane and/or renal proximal tubule after fast absorption in the intestine. In addition, the reduced variation in plasma drug concentrations was observed in the higher dose groups (2 and 5 mg/kg) of the compound.

Thank you for your consideration of this work.

Sincerely,

Hwa Jeong Lee, Ph.D.

Professor

Round 2

Reviewer 1 Report

Authors have answered the reviewer's points.

Author Response

Response to Reviewer 1 Comment

Point 1: Authors have answered the reviewer's points.

Response 1: Thank you for your comments and suggestions again. Our manuscript has been improved more based on your comments.

Sincerely,

Hwa Jeong Lee, Ph.D.

Professor

Reviewer 3 Report

The authors responded to each problem raised by the review and argued their response.

Row 69: K2CO3

Row 111: I think correct should be N-methyl-benzylamine

Some relevant comments to the question posed by the review, like those mentioned below should be also included in the manuscript. I consider important for the readers to understand some relevant issues:

“…some part of structure showed a similarity between FA derivatives and VER, as you commented. Because of the similar part of structure, FA derivatives might show the effect on the reversal of the P-gp-mediated MDR in cancer cells. The enhancement of cytotoxicity and accumulation of anticancer drug, DNM in P-gp overexpressed breast cancer cells was also observed in our studies. Moreover, in silico study, FA showed hydrogen bonding and hydrophobic interaction with transmembrane domain of P-gp (Eur J Pharmacol 2016:786:194) like VER”

“All the synthesized compounds including 3(5c) have been registered as a patent (KR10-2013-0099676) by Prof. Hea-Young Park and colleagues in our group since 2013”

“the dose-dependent decrease in ATPase activity of VER was also observed in our previous study (Eur J Med Chem, 2015:93:237). However, all the values were over 2 (as a ratio of blank), suggesting ATPase stimulating effect of VER. In our previous studies, the lower dose (50 μΜ) of VER showed less DNM accumulation and efflux results (Eur J Pharmacol 2014:723:381) than higher dose (100 μΜ) of VER (Eur J Med Chem, 2015:93:237; J Pharm Pharmacol 2018:70:234). Therefore, we used 100 μΜ VER and FA derivatives to observe P-gp inhibitory activity as much as possible in in vitro studies. According to the results of ATPase activity assay, four FA derivatives may inhibit P-gp activity by depleting ATP as an ATPase stimulator because a ratio of blank was over 2.”

Author Response

Response to Reviewer 3 Comments

Thank you for your comments on our manuscript, again. We have taken your critiques into account for revising the manuscript. Below we provided the point-to-point responses to your comments. In the revised manuscript, our responses are marked as pink highlights.

Point 1: Row 69: K2CO3

Response 1: Thank you for your comment. As followed by your comment, it was corrected as K2CO3.

Point 2:  Row 111: I think correct should be N-methyl-benzylamine

Response 2: Thank you for your comment. As following by your comment, it was corrected.

Point 3:  Some relevant comments to the question posed by the review, like those mentioned below should be also included in the manuscript. I consider important for the readers to understand some relevant issues:

“…some part of structure showed a similarity between FA derivatives and VER, as you commented. Because of the similar part of structure, FA derivatives might show the effect on the reversal of the P-gp-mediated MDR in cancer cells. The enhancement of cytotoxicity and accumulation of anticancer drug, DNM in P-gp overexpressed breast cancer cells was also observed in our studies. Moreover, in silico study, FA showed hydrogen bonding and hydrophobic interaction with transmembrane domain of P-gp (Eur J Pharmacol 2016:786:194) like VER”

“All the synthesized compounds including 3(5c) have been registered as a patent (KR10-2013-0099676) by Prof. Hea-Young Park and colleagues in our group since 2013”

“the dose-dependent decrease in ATPase activity of VER was also observed in our previous study (Eur J Med Chem, 2015:93:237). However, all the values were over 2 (as a ratio of blank), suggesting ATPase stimulating effect of VER. In our previous studies, the lower dose (50 μΜ) of VER showed less DNM accumulation and efflux results (Eur J Pharmacol 2014:723:381) than higher dose (100 μΜ) of VER (Eur J Med Chem, 2015:93:237; J Pharm Pharmacol 2018:70:234). Therefore, we used 100 μΜ VER and FA derivatives to observe P-gp inhibitory activity as much as possible in in vitro studies. According to the results of ATPase activity assay, four FA derivatives may inhibit P-gp activity by depleting ATP as an ATPase stimulator because a ratio of blank was over 2.”

Row 46: Why the scheme is presented in the introduction?

Response 3: Thank you very much for the comments. As followed by your comments, we inserted those mentions in the end of ‘Materials’ and the initial part of ‘Discussion’, respectively. These are pink highlighted.

Thank you for your consideration of this work.

Sincerely,

Hwa Jeong Lee, Ph.D.

Professor

Reviewer 4 Report

The quality of the manuscript is not good enough for publication. Why there is a lack of dose-dependent change in the AUC and Cmax? 

Author Response

Response to Reviewer 4 Comments

Point 1: The quality of the manuscript is not good enough for publication. Why there is a lack of dose-dependent change in the AUC and Cmax

We respect your opinion. Nevertheless, we believe that our work is valuable to publish in Pharmaceutics as a short communication because this is the first observation that the synthesized ferulic acid (FA) derivative enhances the bioavailability of PTX by potentially inhibiting P-gp present in the elimination organs.

Depending on physicochemical and permeability properties of the compound, there could be an optimal dose to provide a maximum P-gp inhibitory activity. In our previous studies, we have also observed the lack of dose-dependent changes in the AUC and Cmax values (Eur J Pharm Sci, 2012:45:296-301; J Pharm Pharmacol, 2018:70:234-241).

For example, in the previous reports, the magnitude of the increases in PK parameters of PTX was decreased with increasing dose of silymarin, a potential P-gp inhibitor due, at least in part, to its low aqueous solubility. Thus, the poor solubility of silymarin may limit its P-gp inhibitory effect in the small intestine at high doses. Another possible explanation is that P-gp inhibition by silymarin can be saturated at high doses (Eur J Pharm Sci, 2012:45:296-301). In the case of benzoxanthone analogue, P-gp inhibition might be saturated by the high dose or limited by the high dose due to its physicochemical and permeability properties (J Pharm Pharmacol, 2018:70:234-241).

In this study, 2 mg/kg of compound 3(5c) was more effective than 5 mg/kg of the compound in terms of PK parameters of PTX. Therefore, 2 mg/kg of compound 3(5c) might be optimal dose for increasing relative bioavailability of PTX. However, because 2 mg/kg of compound 3(5c) did not show statistical significance on the AUCinf of PTX versus control (PTX only) whereas 5 mg/kg of the compound significantly increased the AUCinf (Table 3), we did not claim that 2 mg/kg of compound 3(5c) showed more effects than 5 mg/kg. Rather, we mentioned the changes in PK parameters of PTX at higher doses (2 and 5 mg/kg) as compared with those of 0.5 mg/kg dose in the manuscript.

It has been reported that FA is quickly absorbed in the intestine with 5-15 min and 30 min of Tmax values in rats and humans, respectively, and is mainly excreted through kidney. We think that the interaction between FA derivative, quickly absorbed at GI tract, and PTX on P-gp efflux transporters might occur during the elimination process. Therefore, the compound 3(5c) seemed not to affect absorption of PTX, resulting in no significant change in Cmax value.

Thank you for your consideration of this work.

Sincerely,

Hwa Jeong Lee, Ph.D.

Professor
